# Super-sectioning with multi-sheet reversible saturable optical fluorescence transitions (RESOLFT) microscopy

Andreas Bodén[1], Dirk Ollech [1], Andrew G. York[2], Alfred Millett-Sikking[2] & Ilaria Testa [1] ✉

Light-sheet fluorescence microscopy is an invaluable tool for four-dimensional biological imaging of multicellular systems due to the rapid volumetric imaging and minimal illumination dosage. However, it is challenging to retrieve fine subcellular information, especially in living cells, due to the width of the sheet of light (>1 μm). Here, using reversibly switchable fluorescent proteins (RSFPs) and a periodic light pattern for photoswitching, we demonstrate a super-resolution imaging method for rapid volumetric imaging of subcellular structures called multi-sheet RESOLFT. Multiple emission-sheets with a width that is far below the diffraction limit are created in parallel increasing recording speed (1–2 Hz) to provide super-sectioning ability (<100 nm). Our technology is compatible with various RSFPs due to its minimal requirement in the number of switching cycles and can be used to study a plethora of cellular structures. We track cellular processes such as cell division, actin motion and the dynamics of virus-like particles in three dimensions.

In light-sheet fluorescence microscopy (LSFM)[1] most of the illumination delivered to the sample volume is restricted to the focal plane, which sections the volume into images with reduced background and significantly reduces photodamage. The thickness of the light sheet is important in determining the microscope's ability to resolve spatial features between successive image planes that make up the volume. For example, in traditional LSFM with larger samples (for example, model organisms and organoids), the creation of thin illumination sheets (~2–5 μm) relative to the sample thickness (~>100 μm) is relatively straightforward[1–3] and produces a well-sectioned volume. However, for the investigation of smaller subcellular structures (~1–10 μm) traditional LSFM has limited sectioning ability because the light sheet is relatively thick compared with the features of interest.

Attempts to create thinner light sheets have been made using specialized patterning, photoswitching transitions and two objective lenses facing the sample[4–6]. However, their use in biological time-lapse imaging remains challenging due to inefficient detection of photons,

sample accessibility and sub-optimal photoswitching, which compromises the practical spatiotemporal resolution. LSFM designs based on oblique plane microscopy (OPM)[7–13] solve the challenges of sample accessibility and detection efficiency, and enable imaging in conventional samples such as slides and multiwell plates with a high numerical aperture (NA).

However, due to the diffractive nature of light, even a high-NA OPM system (for example, NA ~ 1.3) has limited sectioning ability. For example, an illumination light sheet that has to propagate through tens of micrometers of sample can never be made thinner than approximately 1–2 μm. Here, we overcome this limitation by utilizing reversible saturable optical fluorescence transitions[14–16] (RESOLFT) and a parallelized switching scheme to excite up to 10-fold thinner slices than previous OPM systems and thereby achieve super-sectioned volumes. The use of reversibly switchable fluorescent proteins (RSFPs)[17,18] is key to achieving the super thin sections. RSFPs have two distinct conformational states, termed the 'on' (fluorescent) and the 'off' (dark) state. What makes

[1]Department of Applied Physics and Science for Life Laboratory, KTH Royal Institute of Technology, Stockholm, Sweden. [2]Calico Life Sciences LLC, South San Francisco, CA, USA. ✉e-mail: ilaria.testa@scilifelab.se

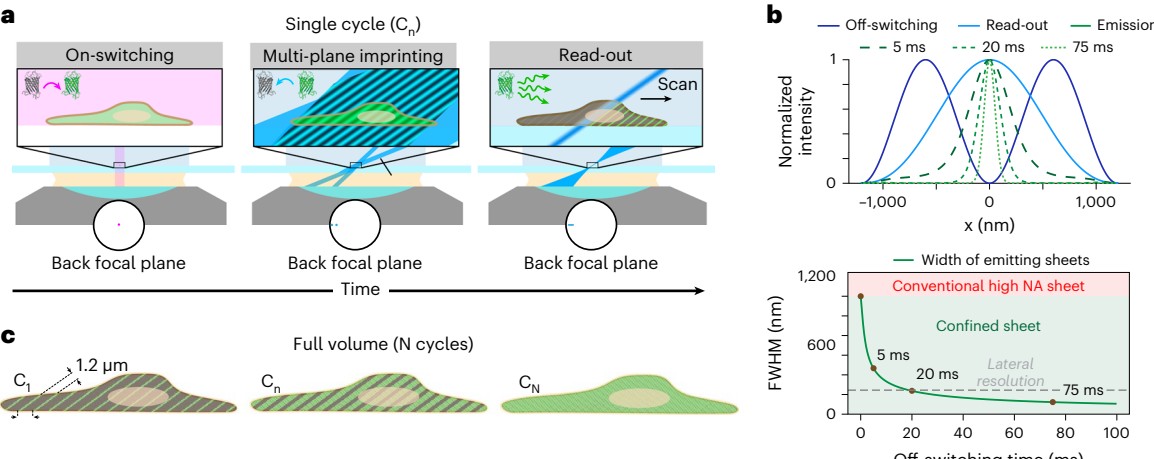

**Fig. 1 | Multi-sheet RESOLFT concept. a,** The recording scheme consists of a widefield on-switching illumination to switch the RSFPs into the on state in the whole cell, an off-switching pulse to confine multiple thin planes of RSFPs in the on state, and a blue sheet of light to excite the RSFPs left in the on state. **b,** The thickness of the confined sheets depends on the time and intensity of the off-switching illumination. The top graph shows color-coded cross-sectional intensity profiles of the off-switching and read-out illuminations and the predicted confinement at 50 W cm⁻² of off-switching intensity for different times with rsEGFP2 labeling. The bottom graph shows the expected thickness of each emitting sheet at increased off-switching times. FWHM, full width at half-maximum. **c,** Each imaging cycle images a subset of confined planes in the sample. To image the full sample the imaging cycle needs to be repeated N times to sample the space between each confined plane.

them especially interesting for optical microscopy is that illumination at different wavelengths and relatively low intensities (W-kW cm⁻²) can enable RSFPs to switch into one of the two states, which is stable over extended times (from seconds to hours). Thus, if a sample labeled with RSFPs is exposed to a structured illumination pulse of a wavelength that induces photoswitching, a pattern of on-state RSFPs can be created in the sample that will remain until it is perturbed again. Higher resolution is achieved by leaving a sub-diffractive pattern of on-state RSFPs in the sample that can be imaged, and then repeating this process with a series of patterns to produce a super-resolved volume.

The multi-sheet RESOLFT microscope is based on the idea of imprinting thin planes of on-state RSFPs extending throughout the full volume to be imaged, and then reading them out quickly (0.4–0.6 kHz) with light-sheet excitation (Fig. 1a). The combination of volumetric imprinting and light-sheet read-out maximizes the signal from the RSFPs and enables the acquisition of volumetric data with high-resolution content in which, importantly, out-of-focus on–off cycling is significantly reduced. This minimizes the number of switching cycles required for imaging. Given that RSFP switching is a limiting factor both in terms of imaging speed and time-lapse duration, minimization of cycling is essential for fast and long-term imaging. With our technique, only around 10–20 switching cycles are needed for a single volume. Leveraging this, we demonstrate super-sectioned volumetric recordings of up to 140 × 84 × 15 μm³, volumetric time-lapse recordings of up to 60 timepoints, and sub-second acquisition times of whole-cell volumes. This enables imaging of subcellular structures across large volumes over multiple timepoints, enabling us to track a plethora of cellular processes such as cell division, cytoskeleton dynamics, and clustering of membrane proteins involved in the formation of virus-like particles.

## Results

### Multi-sheet RESOLFT imaging scheme

The multi-sheet RESOLFT recording scheme (Fig. 1a) starts with a pulse of widefield illumination at 405 nm that switches all of the RSPFs into the on state. Thereafter, the sample is illuminated with off-switching light crafted into a sinusoidal interference pattern with a periodicity of 1.2 μm, which is tilted 35° from the sample plane. This leaves RSFPs in the on state only along tightly confined sheets spanning the sample, the thickness of which can be adjusted by tuning the power and duration of the off-switching pulse and has no theoretical hard limit on its confinement (Fig. 1b). In practice, labeling densities, photoswitching noise and crosstalk enabled us to reach 100–200 nm confinement, which is up to 10-fold smaller than the width of a traditional excited sheet.

When an array of on-state sheets has been created in the sample, a 'read-out' light-sheet is used to excite fluorescence from one confined sheet at a time. The distance between adjacent confined planes is matched to the width of the read-out sheet so that it illuminates only a single confined plane at a time (Fig. 1a, Read-out). This is achieved by matching the patterns in space so that the confined planes adjacent to the one being read out overlap with the first zero-intensity point in the Airy-shaped intensity profile of the read-out light-sheet. Getting this design correct is vital to minimize the crosstalk between adjacent confined planes. Additionally, given that the RSPFs used are negative switchers (meaning that the RSFPs are excited and switched off by the same wavelength), sufficient separation is needed to avoid switching off the out-of-focus confined sheet. The reason for using negative switchers is that, of the existing library of RSFPs, they exhibit the fastest kinetics and withstand the highest number of on–off switching cycles. Positive switchers could potentially also be used but care should be taken to make the read-out pulses sufficiently short so as to not switch on new proteins as this would interfere with the sheet confinement. Future developments may produce switching proteins with spectrally decoupled switching and excitation that could further enhance the performance of the proposed system, but current variants are either too slow in switching or in need of relatively large doses of high-energy light at 350–405 nm.

The imprinting of confined sheets and read-out sweep provides information about a subset of sharply confined two-dimensional sections through the sample. To acquire data on the whole sample volume, this sequence needs to be repeated multiple times, each time shifting the imprinting pattern slightly to acquire a different subset of planes until the distance between adjacent imprinted sheets has been covered (Fig. 1c). At that point, the full volume can be reconstructed with a resolution determined by the optical resolution of the detection (~250 nm) combined with the thickness of the sheets imprinted in the sample (100–200 nm). To explore the potential of the proposed imaging scheme, we performed simulated imaging

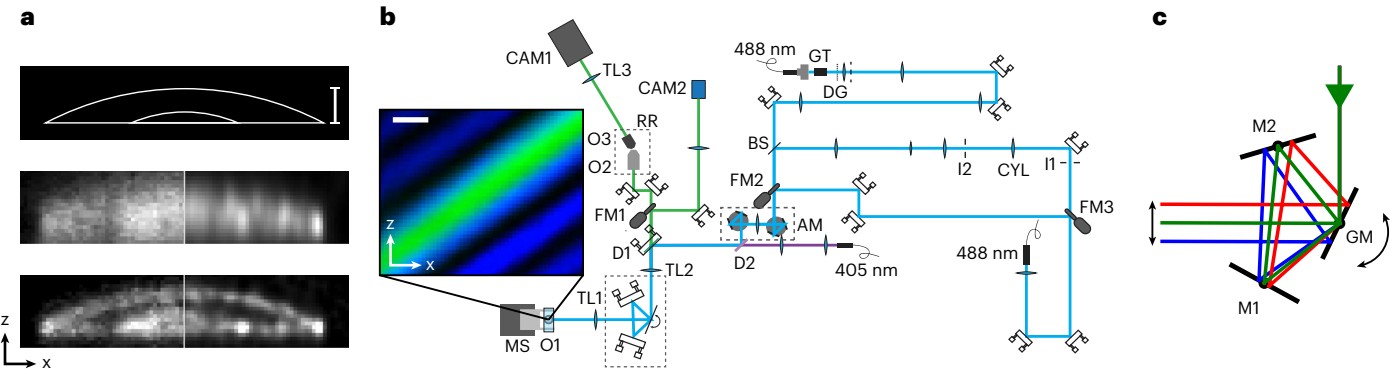

**Fig. 2 | Simulation and optical set-up. a**, Simulated imaging of a virtual sample consisting of three membranes spaced at sub-micrometer distances. The top image shows the ground truth; in the middle and the bottom pictures the imaging is performed with conventional OPM and multi-sheet RESOLFT, respectively. The left side of the images shows reconstruction using the Simple deskew algorithm, while the right side is reconstructed using the Deconvolution deskew algorithm. **b**, Schematic diagram of the optical set-up used for multi-sheet RESOLFT imaging. AM, alignment module; BS, beam splitter; CAM1, oblique detection camera; CAM2, orthogonal detection camera; CYL, cylindrical lens; DG, diffraction grid; $D_N$, dichroic mirror; FM, flip mirror; GM, galvanometric mirror; GT, Glan-Thompson polarizer; $I_N$, iris; $M_N$, mirror; MS, motorized stage; $O_N$, objective; RR, remote refocus module; SM, scanning module; $TL_N$, tube lens. The inset on the left shows the illumination pattern intensities probed with a fluorescent bead and reconstructed. **c**, Schematic diagram of the single galvanometric scan unit used for moving the illuminations and detection with respect to the sample. Scale bars, 1 μm.

(Methods and Supplementary Note 3) on a virtual sample mimicking a multi-membrane structure to compare the achievable resolution when adding the additional sheet confinement (Fig. 2a). We found that the additional confinement lets us resolve and distinguish the different membranes inside the structure that are blurred together when using only the traditional excitation sheet.

The total acquisition time of a full multi-sheet RESOLFT volume depends on several factors. Most importantly, the time is determined by the switching speed of the RSFPs, the length of the scan (that is, the size of the lateral field of view (FOV) in the scanning direction) and the lateral distance between adjacent acquired planes. Another factor that also affects the recording time is the read-out time of the camera, which in turn scales with the axial FOV needed (number of lines to read out on the camera). The fastest recording time reached so far is ~0.7 seconds, and this was achieved over a 57 × 42 × 8 μm³ FOV with a 210 nm lateral distance between acquired planes and using the faster reversibly photoswitchable enhanced green fluorescent protein 2 (rsEGFP2) protein. With the slower rsEGFP(N205S) protein and larger FOV, acquisition times can go up to ~5 seconds for single volumes. Full details on imaging parameters are given in Supplementary Table 1. With higher-power lasers and the development of even faster cameras, we expect that the imaging speed could be pushed significantly below 0.5 seconds for a full volume.

To enable the multi-sheet RESOLFT imaging we built a new optical set-up (Fig. 2b) based on the optical detection design of recently published OPM systems[11], but swapping out the traditional scan section for a more light-efficient lens-free scan system (Methods, Fig. 2c and Supplementary Fig. 2) that achieves a pure beam translation in remote object space by reflecting twice off a large galvanometric mirror[19]. This scan system simplifies the alignment procedure and significantly increases detection efficiency.

Compared with traditional OPM, multi-sheet RESOLFT requires an additional widefield illumination path to generate the on-switching and another one for generating the spatially modulated off-switching illumination. The interference pattern used for off-switching is created by passing a vertically polarized, collimated laser beam through a diffraction grating inducing a periodic 0/π phase pattern. After selecting the first diffraction orders using a physical mask, the beams pass a series of lenses and mirrors to achieve the needed modulation frequency in sample space. The same wavelength at 488 nm is used to generate a light-sheet illumination to excite the RSFPs left in the

on state. Before performing imaging experiments, the structure and alignment of the different illumination patterns were visualized by probing the intensities in the sample using a 200 nm fluorescent bead and using the data to reconstruct an image of the illumination patterns. (Fig. 2b, inset).

### Live cell and time-lapse imaging

To demonstrate the superior sectioning ability of our technology, we imaged living cells transfected with a fluorescently labeled microtubule-associated protein MAP2-rsEGFP(N205S) (Fig. 3 and Extended Data Fig. 1), achieving an effective axial point spread function (PSF) size of <200 nm (Fig. 3b). The whole microtubular network of two cells was captured in a volumetric recording composed of 400 tilted slices covering a volumetric FOV of 90 × 42 × 11 μm³. The cells were imaged without (Fig. 3c) and with (Fig. 3d lower right) the off-switching light to compare multi-sheet RESOLFT and conventional OPM imaging data. When not using the off-switching pulse, the effective excitation thickness corresponds to the width of the read-out light sheet, giving an expected sectioning of between 1 μm and 1.5 μm (Supplementary Fig. 4c). When the confining illumination is applied after the on-switching pulse, the effective excitation thickness is confined, resulting in effective sheet thicknesses below 200 nm (Fig. 1b bottom graph and Fig. 3b). This confinement along the tilted light-sheet direction, together with the lateral resolution of ~250 nm (~1.2 NA) of the optical detection, leads to our technology reaching a final volumetric resolution of at least 250 nm in the x, y and z dimensions (Supplementary Fig. 5).

Multi-sheet RESOLFT can be used to image different dynamic structures and it is compatible with different RSFPs (Figs. 3–6). We performed time-lapse imaging of several structures to observe cellular processes such as cytoskeleton remodeling, cell division, virus particle movements and organelle dynamics. A colony of actin-labeled cells was imaged volumetrically (300 tilted slices, 130 × 63 × 11 μm³) over 30 min (Fig. 4a, Extended Data Fig. 2a and Supplementary Video 1) and a different cell was imaged at a single timepoint (Fig. 4b and Extended Data Fig. 2b). Each volume was recorded in 2 seconds and the time-lapse spans 30 volumes maintaining >75% of the original signal (Fig. 4a, inset graph). We also followed division events of cells expressing the rsEGFP2-labeled histone H2B (Fig. 5 and Supplementary Videos 2 and 3). The recording was started in prometaphase with clearly condensed chromosomes. We monitored the chromosome reorganization,

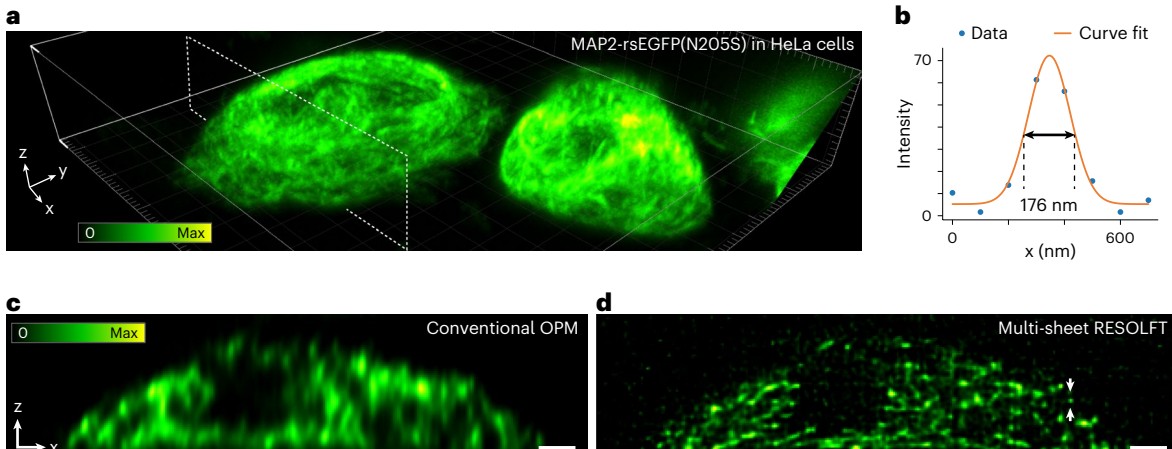

**Fig. 3 | Multi-sheet RESOLFT imaging of a microtubular network.** Imaging of the microtubular network labeled using MAP2-rsEGFP(N205S) transfection. The reconstructions are the average of two sequential volumetric recordings. **a**, Volumetric rendering of the sample with the position of the plane shown in **c** and **d** indicated. **b**, The graph shows a line profile over the isolated filament marked in **d**. Note that the line profile is measured in a non-deconvolved

(Simple deskew) reconstruction (Extended Data Fig. 1). **c**, The axial slice indicated in the volumetric rendering imaged without multi-sheet RESOLFT confinement. **d**, The axial slice indicated in the volumetric rendering imaged with multi-sheet RESOLFT confinement. All images have been reconstructed using the Deconvolution deskew algorithm (Methods). Scale bars, 2 μm.

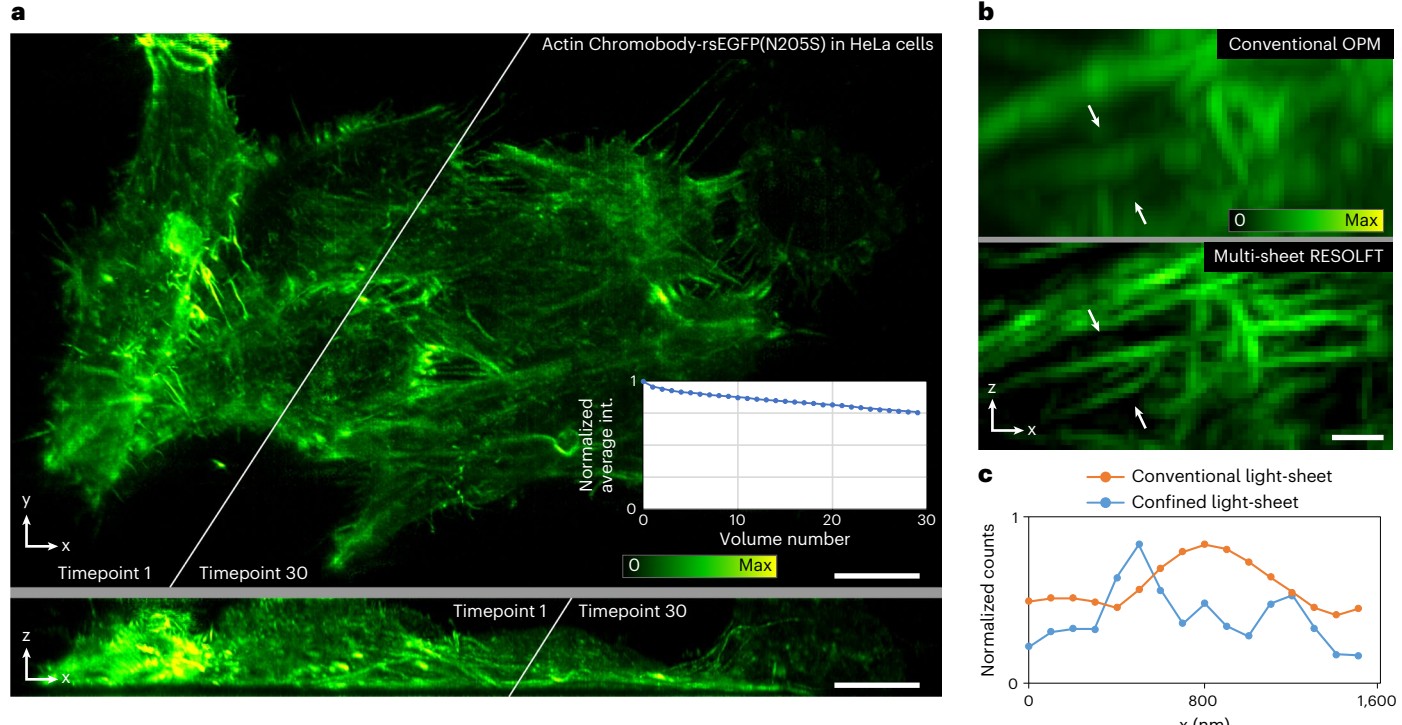

**Fig. 4 | Multi-sheet RESOLFT time-lapse imaging of actin cytoskeleton. a**, HeLa cells labeled with Actin-Chromobody-rsEGFP2. A larger colony of cells is recorded volumetrically over 29 min at 1 volume min⁻¹ shown as a maximum intensity projection along $z$ (top) and $y$ (bottom). The inset shows the maintained average signal intensity (int.) throughout the time-lapse recording. **b**, A single timepoint

of conventional OPM data and multi-sheet RESOLFT data from a small region of a different cell as a maximum intensity projection along $y$. **c**, The line profiles show distinguishable filaments in the multi-sheet RESOLFT data that are non-resolvable in the conventional OPM modality. All images have been reconstructed using the Deconvolution deskew algorithm (Methods). Scale bars: **a**, 10 μm; **b**, 1 μm.

segregation and de-condensation during the mitotic process, and chromatin dynamics in the interphasic nuclei of the resulting daughter cells for more than 4 hours (20 frames with 15 minute intervals), confirming the live cell compatibility of our method.

Thanks to the improved resolution in sequential volumetric time-lapse recordings, we could capture rare stochastic events in various subcellular regions simultaneously. As an example, we recorded

the entire mitochondrial network of a cell with 60 timepoints over 2 minutes and detected mitochondrial fission and fusion events at the perinuclear region and cell periphery (Supplementary Video 4). The video also demonstrates the high imaging speed of the system by acquiring the full volumetric timepoints in ~700 ms.

Another advantage of volumetric recordings is the possibility to simultaneously capture the entire cell membrane, which, combined

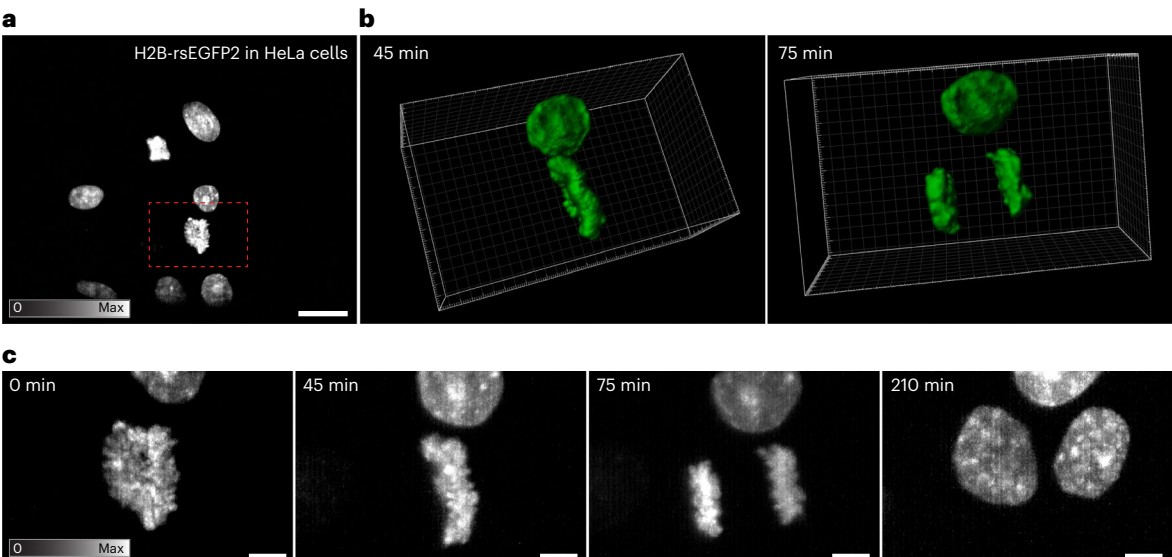

**Fig. 5 | Multi-sheet RESOLFT time-lapse imaging of cell division.** HeLa cells labeled with H2B-rsEGFP2 were imaged every 15 min for 4 h and 45 min. **a**, The full FOV of the recording is shown with a red square indicating the smaller FOV shown in **b** and **c**. **b**, Two frames taken from a volumetric rendering of the time-lapse recording (Supplementary Video 3). **c**, Four selected frames from the maximum intensity projection time-lapse showing significant steps in the cell division process. All images have been reconstructed using the Simple deskew algorithm (Methods). Scale bars: **a**, 20 µm; **c**, 5 µm.

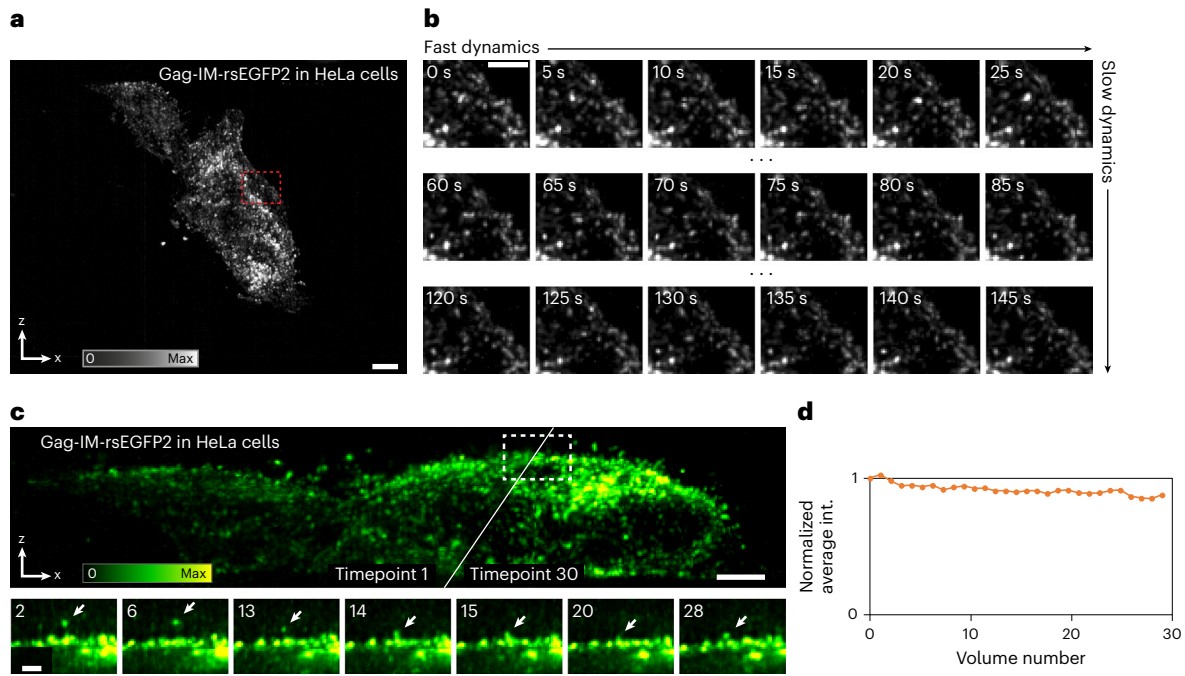

**Fig. 6 | Multi-sheet RESOLFT time-lapse imaging of virus-like particles.** Time-lapse imaging of HeLa cells expressing the Gag-IM-rsEGFP2 fusion protein forming virus-like particles, mainly at the cell membrane (IM, intra-molecular). **a**, Maximum intensity projection over both time and the $z$ dimension of the time-lapse recording. **b**, The three rows show three different cropped six-frame sequences taken from the first, middle and last part of the whole recording. Each image is a maximum intensity projection along $z$ from a cropped volume as indicated in **a**. **c**, The top row data show the same recording as in **a** and **b** but as the maximum intensity projection of the first and last frame into the $x$–$z$ plane. The bottom row shows zoom-ins of the marked region at seven different timepoints, showing a particle moving around just above the apical membrane of the cell. These frames are displayed as the maximum intensity projection of a small $5.5 \times 4 \times 1.6$ µm³ ($x,z,y$) volume of interest around the identified particle. **d**, Graph of the maintained average fluorescence intensity during the time-lapse recordings shown in **a** and **b**. All images have been reconstructed using the Deconvolution deskew algorithm (Methods). Scale bars: **a**, 5 µm; **b**, 2 µm; **c**, 3 µm (top), 1 µm (bottom).

with the high spatiotemporal resolution, enabled us to follow the dynamics of virus-like particles at the apical membrane side of the cell (Fig. 6, Extended Data Figs. 3, 4 and Supplementary Video 5) (300 tilted slices, $80 \times 42 \times 19$ µm³). HeLa cells expressing an HIV Gag-IM-rsEGFP2

fusion protein show clusters of these fusion proteins at the cell membrane, resulting in the formation of viral-like particles, which could be followed over time. Previous live studies of HIV particle formation and budding are usually restricted to the basal membrane, where

a sufficient axial resolution can be achieved in TIRF (total internal reflection fluorescence) mode illumination[20]. The full set of acquisition parameters for all presented images is given in Supplementary Table 1.

## Discussion

We have developed a new multi-sheet RESOLFT microscope for high-speed (>1 Hz) volumetric imaging (~100 × 80 × 15 µm³) of subcellular structures with super-resolved sectioning (demonstrated here with spatial resolution below 250 nm in $x$, $y$ and $z$). Using a single primary objective for both illumination and detection, the fast volumetric imaging is compatible with standard coverslips, paving the way for high-throughput, high-resolution, volumetric live cell imaging.

Current light-sheet microscopes using conventional fluorophores exhibit limited optical sectioning due to the inherent trade-off between the thickness and the propagation length of the illumination sheet and are therefore limited to optical sectioning of ~1–1.5 µm if a reasonable light-sheet propagation is to be maintained. Confining the light-sheet thickness using the RESOLFT principle, however, has resulted in severely compromised image acquisition times[4].

In multi-sheet RESOLFT, the degree of sectioning can be tuned with more or less off-switching, and adjustment of light-sheet width, step size and so on according to the desired spatial resolution and signal-to-noise ratio.

The improved spatiotemporal resolution of this technology was achieved thanks to the volumetrically parallelized imprinting of multiple thin sheets. This enables a full volumetric recording to be acquired with only 10–20 RSFP switching cycles, minimizing both recording time and light-induced photo-toxicity.

Due to the minimal requirement in cycle number, multi-sheet RESOLFT can be adapted to other RSFPs[21–25], enabling multi-species detection either with kinetics[18] or spectral separation[26].

When imaging RSFPs it is important to match the correct light doses during photoswitching and fluorescence excitation to avoid unwanted photobleaching and low image quality. For example, an excess of 405 nm light during on-switching can cause drastic photobleaching.

Three-dimensional cultures, expanded samples and multicellular organisms can benefit from the new trade-off in volumetric speed and spatial resolution of multi-sheet RESOLFT. However, special care should be taken in preserving the reversible photoswitching during expansion protocols or to compensate for potential aberration of the sheet when illuminating scattering tissues.

Given that present-day RSFPs can be cycled on average up to 1,000–2,000 cycles, our method can also be combined with additional modulated patterns and image rotators[27] to further push the spatial resolution in the focal plane. Oil objective lenses and remote refocus correction can be implemented to achieve higher frequency patterns for off-switching[28]. Finally, photoswitching can also be a valuable tool to increase the sheet homogeneity while maintaining relatively high NA for detection, which can facilitate multiscale[29] investigation of large samples with tunable spatial resolution, especially when coupled with smart automated recording strategies[30,31].

## Online content

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

## Methods

### Simulations

The simulated data shown in Fig. 2a are generated using a custom-written computational tool that can simulate a wide range of imaging schemes. The simulations realistically represent fluorophore behavior by incorporating information about fluorescence cross-section, RSFP switching kinetics, absorption spectra for the different transitions and fluorescence emission spectra. The illumination patterns are represented as three-dimensional intensity distributions and the optical imaging is represented as a convolution with a three-dimensional PSF and a geometrical transform corresponding to a potentially oblique detection. The imaged intensity is then passed through a virtual emission filter and detected by a camera with defined camera properties such as pixel size, quantum efficiency, conversion factor and read-out noise.

The virtual sample used for a simulated imaging experiment is created by distributing discrete fluorophores at positions corresponding to a certain sample geometry. An imaging sequence is then defined and applied that contains illumination pulses of predefined illumination patterns, scanning steps and camera exposures potentially mimicking a wide range of imaging schemes.

The GPU (graphics processing unit)-accelerated computational parts are packaged in a graphical user interface to enable improved ease of use.

### Optical set-up

The multi-sheet RESOLFT microscope is built as a bespoke system in an optical lab. All optical components are commercially available apart from the diffraction grating used in generating the off-pattern (for details contact the authors). All lenses detailed below are aligned so that the focal planes of two adjacent lenses overlap (the so-called 4f configuration). Abbreviations of optical elements correspond to the notations in the schematic diagrams of the set-up.

The multi-sheet RESOLFT microscope uses a Nikon 1.35 NA 100× silicone objective (O1, CFI SR HP Plan Apo Lambda S 100XC Sil) together with a 200 mm tube lens (TL1, Thorlabs TTL-200-A) as the primary magnifying unit. The primary objective is placed underneath the sample, which is held by a 3-Axis NanoMax Stage with stepper motor actuators. Following the detection path, light passes through the scan module (SM) consisting of two fixed 2 inch mirrors and a 50-mm-wide galvanometric mirror (ScanLab dynAXIS 3L with analog amplifier board SSV30) arranged as shown in Fig. 2c. The light then passes through a 358 mm tube lens (TL2, Applied Scientific Instrumentation, LENS-358-A), the primary dichroic mirror (D1, Semrock Di03-R488-t3-25×36) and is then directed by two additional mirrors into the secondary Nikon 0.95 NA 40× air objective (O2, Nikon CFI Plan Apochromat Lambda D 40× 0.95 NA). The remote image created by the secondary objective is then imaged at a 35° angle by the tertiary bespoke 1.0 NA 40× glass-tipped objective (O3, Applied Scientific Instrumentation, AMS-AGY v1). The last optical element in the detection path is a tertiary tube lens (TL3) that focuses the image onto the camera (CAM1). During the experimental imaging, two different cameras were used with pixel sizes of 4.6 μm and 6.5 μm, respectively (Hamamatsu ORCA-Quest and Hamamatsu ORCA-Fusion). To maintain a suitable effective pixel size, a 165 mm tertiary tube lens was used with the ORCA-Quest camera to give a total magnification of 46.1× and an effective pixel size of 100 nm. When using the ORCA-Fusion camera, a 200 mm tertiary tube lens was used, giving a total magnification of 55.9× and an effective pixel size of 116 nm.

The oblique light sheet is generated using a fiber-coupled 488 nm diode laser (HÜBNER Photonics Cobolt 06-MLD 488 nm 200 mW) that is collimated using a 100 mm lens (Thorlabs AC254-100-A-ML). After two mirrors the beam passes through a mechanical slit (I1, Thorlabs VA100CP) used for adjusting the light-sheet width in the sample. After another mirror, the beam passes through a 200 mm cylindrical

lens (CYL, Thorlabs LJ1653RM-A). A second identical mechanical slit (I2) rotated 90° from the first one then enables adjustment of the effective NA of the light sheet by cropping the length of the line that is formed on the back focal plane. The beam is then de-magnified using a telescope consisting of an 80 mm and a 19 mm lens (Thorlabs AC254-080-A-ML and AC127-019-A-ML). After a 200 mm relay lens (Thorlabs AC254-200-A-ML) the path is combined with the off-switching path using a 50:50 non-polarizing plate beam splitter (BS, Thorlabs BSW10).

The off-switching pattern also uses a fiber-coupled 488 nm diode laser (HÜBNER Photonics Cobolt 06-MLD 488 nm 200 mW) that is collimated using a 5 mm laser beam coupler (Schäfter + Kirchhoff 60SMS-1-4-M5-33) and then passed through a Glan-Thompson polarizer (GT, Thorlabs GTH10M-A) to ensure vertical linear polarization. The beam then hits a diffraction grating with a periodic 0/π phase pattern (DG, phase-diffraction grating consisting of 437-nm-high SiO$_2$ lines with a 10 μm period from Laser Laboratorium Göttingen). The diffracted beams are focused with an 8 mm lens (Thorlabs AC080-020-A-ML) onto a physical mask that allows only the +1 and −1 diffraction order to pass through. For geometrical reasons and to adjust the magnification, the beam then passes through a telescope consisting of a 125 mm and a 400 mm lens (Thorlabs AC254-125-A-ML and AC245-400-A-ML) before passing through the 50:50 non-polarizing plate beam splitter (BS) that combines the path with the light-sheet path.

After the 50:50 plate beam splitter, the beams reflect off the first of two beam translation units that form the alignment module (AM). These consist of two 1 inch mirrors mounted on top of a manual rotation stage (Supplementary Fig. 2). As the stage with the mirrors is rotated, the output beam undergoes a pure lateral translation. This translation is used to align the illumination patterns to the focal plane of the oblique optical detection. Between the two beam translation units is a 200 mm lens (Thorlabs AC245-200-A-ML). The translation units are in conjugate spaces, meaning that one of the translation units can laterally shift the illumination patterns in the sample while the other one can adjust the angle of the patterns in the sample. After the second translation unit, the beams reflect off a short-pass dichroic mirror (D2, Thorlabs DMSP425) used to couple the 405 nm beam path to the main illumination path.

The on-switching widefield illumination is created using a fiber-coupled 405 nm diode laser (HÜBNER Photonics Cobolt 06-MLD 405 nm 150 mW). The output beam from the fiber is collimated using a 100 mm lens (Thorlabs AC245-100-A-ML) and then coupled to the main illumination path using the aforementioned short-pass dichroic mirror (D2).

The three illumination lasers will then reflect off the primary dichroic (D1) and pass through the second tube lens (TL2), the scan module (SM), and the primary tube lens (TL1) before hitting the back focal plane of the primary objective (O1).

### Hardware control

To run and use the developed microscope we use a version of the microscope control software ImSwitch[32] that enables easy and modular interfacing with the many hardware components of the system and provides a user-friendly graphical interface. On top of the USB and RS232-based communication with the cameras, lasers, flip-mirrors and motorized stage handled through ImSwitch architecture, some components of the microscope require fast and precise control during acquisition to achieve the necessary synchronization required by the imaging scheme. For this, a Triggerscope 4.0 (Advanced Research Consulting) is used to send digital TTL (transistor–transistor logic) signals to the lasers and the camera while controlling the galvanometric scanning mirror using analog output from the same device. Communication with the Triggerscope to set parameters and start acquisition is also done using a custom RS232-based widget in ImSwitch.

## Volume reconstruction

To visualize the acquired data as fluorophore density volumes, the intensities recorded with the camera need to be mapped back into the sample space coordinate system. In our imaging pipeline, this can be done in two different algorithms. The first one, Simple deskew, performs a simple geometrical transform followed by a normalized Gaussian interpolation from the camera space to the sample space.

The second reconstruction algorithm, Deconvolution deskew, adds a Richardson–Lucy deconvolution step to the geometrical transform[33,34]. The image formation model used in the iterative deconvolution incorporates the transform corresponding to the oblique geometry, preceded by a convolution with a kernel calculated from knowledge of the sheet confinement given by the RSFP switching kinetics together with the applied illumination scheme and the PSF of the optical detection path.

Both algorithms have been implemented in Python with GPU-accelerated computing using Cupy and Numba. If reconstructing on a 100 nm voxelated grid, the Simple deskew algorithm reconstructs a standard volumetric acquisition (~50 × 50 × 15 µm³, 210 nm lateral scan step) in around 2 s and the Deconvolution deskew performs a full 10 iteration volumetric deconvolution reconstruction in around 25 s. Tests were run using a Dell Precision 7820 Tower with an Intel(R) Xeon(R) Silver 4208 CPU with a 2.10 GHz processor, 64 GB RAM and an NVIDIA RTX A4000 GPU.

## Samples preparation

HeLa (ATCC CCL-2) cells were cultured in DMEM (Thermo Fisher Scientific, cat. no. 41966029) supplemented with 10% (v/v) fetal bovine serum (FBS, Thermo Fisher Scientific, cat. no. 10270106), 1% (v/v) penicillin–streptomycin (Sigma-Aldrich, cat. no. P4333) and kept at 37 °C with 5% $CO_2$ in a humidified incubator. For transfection, 18 mm round coverslips were placed in a 12-well plate and seeded with $4 \times 10^4$ cells per well 1 day before transfection using FuGENE (Promega, no. E2311) following the manufacturer's instructions. At 16–24 h post-transfection the cells were placed in a chamber with phenol-red-free Leibovitz's L-15 Medium (Thermo Fisher Scientific, cat. no. 21083027) supplemented with FBS and penicillin–streptomycin for imaging. The following plasmids were used for exogenous expression: MAP2-rsEGFP(N205S), Actin-Chromobody-rsEGFP(N205S) and Gag-IM-rsEGFP2. H2B-rsEGFP2 was stably expressed in HeLa cells after transfection with pDOS066 (ref. 35) followed by selection with 1 mg ml⁻¹ geneticin (Thermo Fisher Scientific, cat. no. 10131035) and FACS sorting based on rsEGFP fluorescence intensity.

## Cloning of Gag-IM-rsEGFP2.

All enzymes and buffer components were purchased from NEB, DNA-primers were purchased from IDT. rsEGFP2 was intra-molecularly inserted close to the carboxy terminus of the matrix domain of the HIV Gag polyprotein flanked by glycine-rich linkers according to Müller et al.[36]. Due to repetitive DNA sequences in the targeted region and also missing restriction sites, the target plasmid pspAX2 was divided into multiple fragments with overlapping 5′ and 3′ sequences for ligation via Gibson Assembly. Fragments 1 and 2 were generated via restriction enzyme digest of pspAX2 using ApaI and NheI or AgeI and SfoI, respectively, and purified via gel extraction. Fragments 3 and 5 were amplified from the pspAX2 template introducing the desired linker sequences via polymerase chain reaction (PCR) using primers fw3: tttgtcccaaatctg tgcgg and rev3: ctcctcgcccttgctcaccataccaccgatgctaccctgattgct gtgtcctgtgtcagc, or fw4: cggcatggacgagctgtacaagggtggcagcat tgtcagccaaaattaccctatagtgc and rev4: ttggtgtccttccttttccacatttccaac agcc, respectively. Fragment 4 coding for rsEGFP2 was PCR-amplified from IT006 rsEGFP2-Omp25 (ref. 37) using the primers fw4: gtgagc aagggcgaggag and rev4: cttgtacagctcgtccatgccg. Sequencing of the assembled construct confirmed the correct insertion of the rsEGFP2 into gag, but the loss of the pol coding region. Thus, the assembled construct and the original pspAX2 plasmid were digested with EagI and SphI. The large fragment from pspAX2 and the small gag-rsEGFP2 fragment from the new construct were separated via gel extraction and ligated using the Instant Sticky-end Ligase Master Mix. The correct assembly of all ligation points was verified via Sanger sequencing.

MAP2-rsEGFP(N205S) and Actin-Chromobody-rsEGFP(N205S) were a kind gift from S. W. Hell and S. Jakobs (MPI-BCP Göttingen, Germany). pspAX2 was a gift from D. Trono (Addgene plasmid no. 12260; http://n2t.net/addgene:12260; RRID:Addgene_12260).

## Reporting summary

Further information on research design is available in the Nature Portfolio Reporting Summary linked to this article.

## Data availability

All of the raw data for the imaging experiments presented are available at Zenodo https://doi.org/10.5281/zenodo.10474256 (ref. 38). Due to space limitations, all of the reconstructed volumes could not be published, but are available from the authors upon reasonable request.

## Code availability

All the code used for hardware control and image reconstruction is available in its exact versions at https://github.com/TestaLab/Multi-sheet-RESOLFT_code. The source code for the software used for the simulations is currently not publicly accessible, but its working principles are described in detail in Supplementary Note 3. The hardware control software is based on the open-source project ImSwitch, hosted at https://github.com/ImSwitch/ImSwitch.

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

## Acknowledgements

The authors thank G. Marin Aguilera and A. Papalini for measuring the RSFPs photoswitching kinetics. I.T. thanks the CZI (iNano Dynamic Imaging grant) and the Swedish Foundation for Strategic Research (Project no. SAB 19-0033 and FLL15-0031) for supporting the research.

## Author contributions

I.T. and A.B. conceived the idea. I.T. designed and supervised the research. A.B. built the multi-sheet RESOLFT set-up, performed the experiments and analyzed the data. D.O. provided biological

guidance, carried out the cloning and cell culture-related work and imaged the cells. A.G.Y. and A.M.-S. contributed to conceptualization, alignment pipelines and discussion. I.T. and A.B. wrote the paper with contributions from all of the authors.

## Funding

## Competing interests

The authors declare no competing interests.

## Additional information

**Extended data** are available for this paper at https://doi.org/10.1038/s41592-024-02196-8.

**Correspondence and requests for materials** should be addressed to Ilaria Testa.

| Conventional OPM | Multi-sheet RESOLFT |

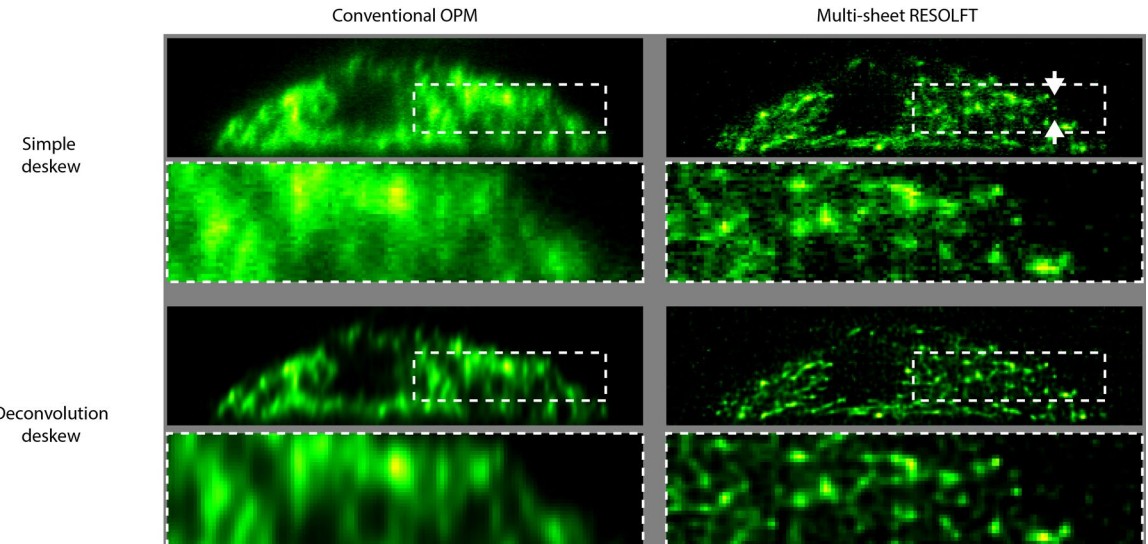

**Extended Data Fig. 1 | Comparison of different OPM modalities and reconstructions.** HeLa cell images presented in Fig. 3c-d shown in full as both conventional OPM data and multi-sheet RESOLFT data and using both the simple deskew reconstruction and the deconvolution deskew with 10 iterations and using the prior of a 1200 nm thick sheet in the conventional OPM data and a 200 nm thick sheet in the multi-sheet RESOLFT data. Arrows in top right indicate where the line profile shown in Fig. 3b is taken. As in Fig. 3c-b, the raw data is the average of two sequential volumetric acquisitions.

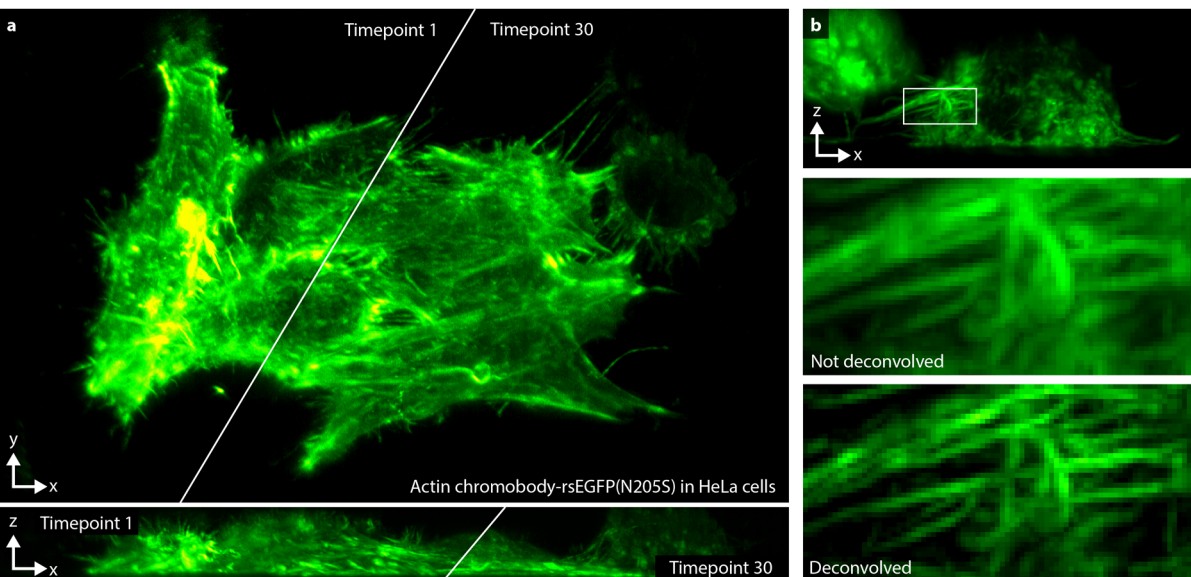

**Extended Data Fig. 2 | Actin chromobody-rsEGFP(N205S) in HeLa cells, non-deconvolved. a**, The same data as shown in Fig. 4a is shown, but reconstructed using the simple deskew algorithm. **b**, The same zoom-in as shown in Fig. 4b (different cell from **a**) is reconstructed and shown using both the simple deskew (*top*) and the algorithm incorporating deconvolution (*bottom*).

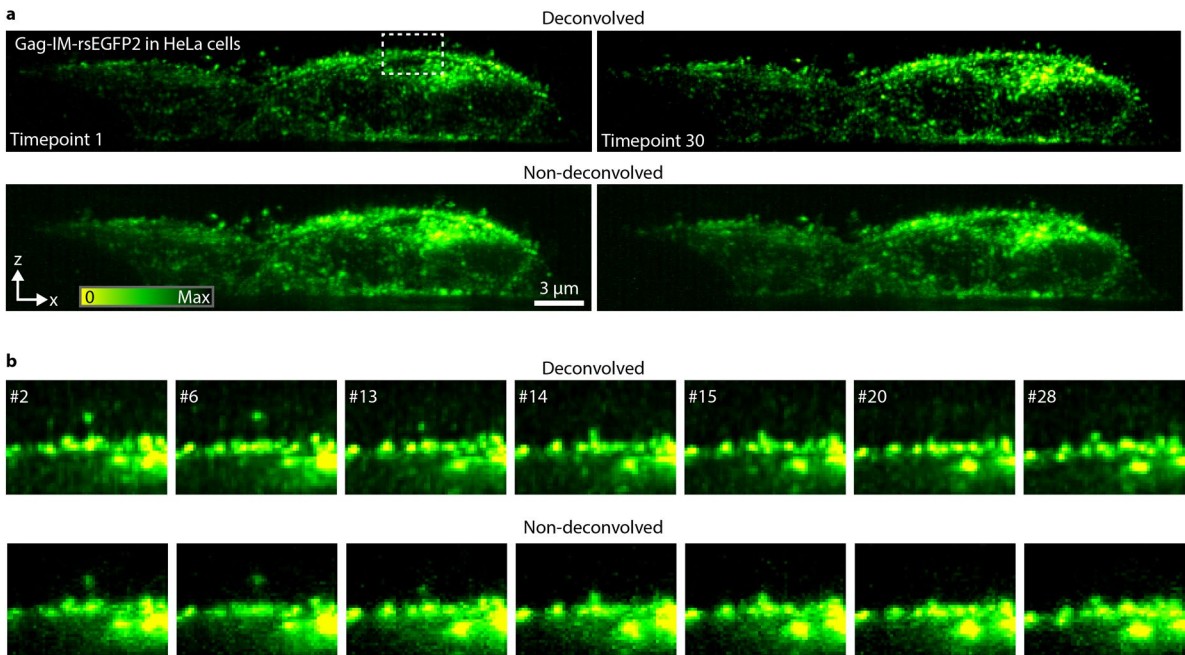

**Extended Data Fig. 3 | Gag-IM-rsEGFP2 in HeLa, reconstruction comparison.**
**a**, The panel shows a comparison between the data reconstructed using the algorithm incorporating a deconvolution step (*top*) and the simple deskew algorithm (*bottom*). Images are maximum intensity projections along the y-direction of the cells. **b**, The same time-lapse points and FOV as shown at the bottom of Fig. 6c are here shown again with the same views taken also from the deconvolved data (*top*) and from the data reconstructed with the simple deskew algorithm (*bottom*).

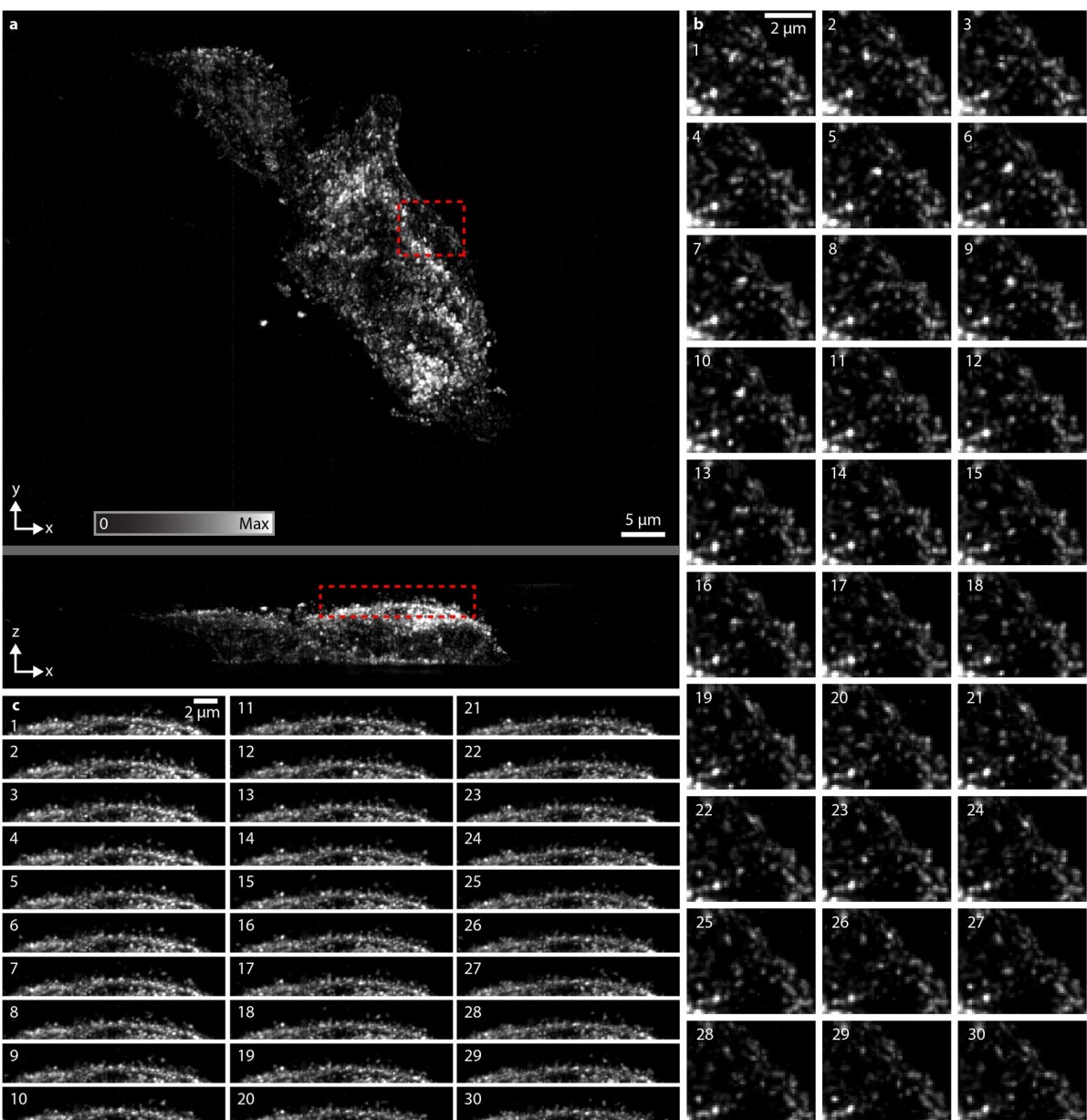

**Extended Data Fig. 4 | Extended data on time-lapse of HeLa cells expressing Gag-IM-rsEGFP2.** Extended data from the acquisition presented in Fig. 6. **a**, Maximum intensity projection over both time and z (top) and time and y (bottom) axis of the 4D data set. **b**, All 30 timepoints of the zoom-in shown in the top panel of **a** as maximum intensity projections along a small section of the z-axis. **c**, All 30 timepoints of the zoom-in shown in the bottom panel of **a** as maximum intensity projections along a small section of the y-axis.

# Reporting Summary

## Statistics

For all statistical analyses, confirm that the following items are present in the figure legend, table legend, main text, or Methods section.

| n/a | Confirmed | |
|---|---|---|
| ☐ | ☒ | The exact sample size (*n*) for each experimental group/condition, given as a discrete number and unit of measurement |
| ☐ | ☒ | A statement on whether measurements were taken from distinct samples or whether the same sample was measured repeatedly |
| ☒ | ☐ | The statistical test(s) used AND whether they are one- or two-sided *Only common tests should be described solely by name; describe more complex techniques in the Methods section.* |
| ☒ | ☐ | A description of all covariates tested |
| ☒ | ☐ | A description of any assumptions or corrections, such as tests of normality and adjustment for multiple comparisons |
| ☐ | ☒ | A full description of the statistical parameters including central tendency (e.g. means) or other basic estimates (e.g. regression coefficient) AND variation (e.g. standard deviation) or associated estimates of uncertainty (e.g. confidence intervals) |
| ☒ | ☐ | For null hypothesis testing, the test statistic (e.g. *F*, *t*, *r*) with confidence intervals, effect sizes, degrees of freedom and *P* value noted *Give P values as exact values whenever suitable.* |
| ☐ | ☒ | For Bayesian analysis, information on the choice of priors and Markov chain Monte Carlo settings |
| ☒ | ☐ | For hierarchical and complex designs, identification of the appropriate level for tests and full reporting of outcomes |
| ☒ | ☐ | Estimates of effect sizes (e.g. Cohen's *d*, Pearson's *r*), indicating how they were calculated |

*Our web collection on statistics for biologists contains articles on many of the points above.*

## Software and code

Policy information about availability of computer code

| Data collection | The data was acquires using a hardware control software based heavily on the ImSwitch architechture. This exact software version used, together with the firmware used on the Triggerscope 4.0 can be found on https://github.com/TestaLab/Multi-sheet-RESOLFT_code. Dependencies and version numbers are listed in each repository. |
|---|---|
| Data analysis | The image reconstruction was done using a custom written reconstruction and deconvolution software that is available at https://github.com/TestaLab/Multi-sheet-RESOLFT_code. Dependencies and version numbers are listed in each repository. The source code for the software used for the simulations is currently not publicly accessible, but its working principles are described in detail in Supplementary Note 3. |

For manuscripts utilizing custom algorithms or software that are central to the research but not yet described in published literature, software must be made available to editors and reviewers. We strongly encourage code deposition in a community repository (e.g. GitHub). See the Nature Portfolio guidelines for submitting code & software for further information.

## Data

Policy information about availability of data

All manuscripts must include a data availability statement. This statement should provide the following information, where applicable:
- Accession codes, unique identifiers, or web links for publicly available datasets
- A description of any restrictions on data availability
- For clinical datasets or third party data, please ensure that the statement adheres to our policy

> All the data presented is available as both raw dataset and reconstructed volumes at Zenodo https://doi.org/10.5281/zenodo.10474256. Due to space limitations, all the reconstructed volumes could not be published, but are available from the authors upon reasonable request.

## Research involving human participants, their data, or biological material

Policy information about studies with human participants or human data. See also policy information about sex, gender (identity/presentation), and sexual orientation and race, ethnicity and racism.

| | |
|---|---|
| Reporting on sex and gender | N/A |
| Reporting on race, ethnicity, or other socially relevant groupings | N/A |
| Population characteristics | N/A |
| Recruitment | N/A |
| Ethics oversight | N/A |

Note that full information on the approval of the study protocol must also be provided in the manuscript.

# Field-specific reporting

Please select the one below that is the best fit for your research. If you are not sure, read the appropriate sections before making your selection.

☒ Life sciences        ☐ Behavioural & social sciences        ☐ Ecological, evolutionary & environmental sciences

For a reference copy of the document with all sections, see nature.com/documents/nr-reporting-summary-flat.pdf

# Life sciences study design

All studies must disclose on these points even when the disclosure is negative.

| | |
|---|---|
| Sample size | The study does not present any data based on statistical analysis and no sample sizes were therefore chosen. Images presented are on individual samples and line-profiles measured are on individual filaments or puncta structures. Imaging performance is demonstrated in multiple different sample types (N = 3) imaged on different days in order to demonstrate the techniques versatility. |
| Data exclusions | No data were excluded from the analysis. |
| Replication | After system alignment recordings could be performed over the coarse of a full day without notable deterioration in image quality. After closing down the system and starting up the following day or multiple days after, minor realignments were sometimes needed. Imaging was performed inside a time period of half a year. All attempts at replication were successful. |
| Randomization | No allocation into experimental group were performed. |
| Blinding | No allocation into experimental group were performed. |

# Reporting for specific materials, systems and methods

We require information from authors about some types of materials, experimental systems and methods used in many studies. Here, indicate whether each material, system or method listed is relevant to your study. If you are not sure if a list item applies to your research, read the appropriate section before selecting a response.

## Materials & experimental systems

| n/a | Involved in the study |
|---|---|
| ☒ | ☐ Antibodies |
| ☐ | ☒ Eukaryotic cell lines |
| ☒ | ☐ Palaeontology and archaeology |
| ☒ | ☐ Animals and other organisms |
| ☒ | ☐ Clinical data |
| ☒ | ☐ Dual use research of concern |
| ☒ | ☐ Plants |

## Methods

| n/a | Involved in the study |
|---|---|
| ☒ | ☐ ChIP-seq |
| ☒ | ☐ Flow cytometry |
| ☒ | ☐ MRI-based neuroimaging |

## Eukaryotic cell lines

Policy information about cell lines and Sex and Gender in Research

| | |
|---|---|
| Cell line source(s) | Hela, ATCC CCL-2 |
| Authentication | None of the cell lines used were authenticated. |
| Mycoplasma contamination | No testing for mycoplasma contamination. |
| Commonly misidentified lines (See ICLAC register) | No commonly misidentified cell lines were used in this study. |

