## [Peer Review File · Nature Methods]

Peer Review Information

Manuscript Title: Super-sectioning with Multi-sheet REversible Saturable Optical Fluorescence Transitions (RESOLFT) microscopy

Corresponding author name(s): Ilaria Testa

Editorial Notes: None

Reviewer Comments & Decisions:

Decision Letter, initial version:

Dear Ilaria,

Your Brief Communication, "Super-sectioning with Multi-sheet REversible Saturable Optical Fluorescence Transitions (RESOLFT) microscopy", has now been seen by three reviewers. As you will see from their comments below, although the reviewers find your work of considerable potential interest, they have raised a number of concerns. We are interested in the possibility of publishing your paper in Nature Methods, but would like to consider your response to these concerns before we reach a final decision on publication. We therefore invite you to revise your manuscript to address these concerns.

We think the referee concerns are largely constructive and fair, and addressing them will help potential users better evaluate the method. We also wanted to note that we think the impact of the work would be enhanced by the addition of imaging on a more challenging, multicellular, sample. We appreciate that this may not be easy to do without specialized biology expertise, so we do not require it. But we do ask that you at least describe the limits of the technology in terms of specimens that can be observed at these resolutions.

* include a point-by-point response to the reviewers and to any editorial suggestions

- * please underline/highlight any additions to the text or areas with other significant changes to facilitate review of the revised manuscript
- * address the points listed described below to conform to our open science requirements
- * ensure it complies with our general format requirements as set out in our guide to authors at www.nature.com/naturemethods
- * resubmit all the necessary files electronically by using the link below to access your home page

[Redacted] This URL links to your confidential home page and associated information about manuscripts you may have submitted, or that you are reviewing for us. If you wish to forward this email to co-authors, please delete the link to your homepage.

We hope to receive your revised paper within three months. If you cannot send it within this time, please let us know. In this event, we will still be happy to reconsider your paper at a later date so long as nothing similar has been accepted for publication at Nature Methods or published elsewhere.

OPEN SCIENCE REQUIREMENTS

REPORTING SUMMARY AND EDITORIAL POLICY CHECKLISTS

DATA AVAILABILITY

We strongly encourage you to deposit all new data associated with the paper in a persistent repository where they can be freely and enduringly accessed. We recommend submitting the data to discipline-specific and community-recognized repositories; a list of repositories is provided here:

<http://www.nature.com/sdata/policies/repositories>

All novel DNA and RNA sequencing data, protein sequences, genetic polymorphisms, linked genotype and phenotype data, gene expression data, macromolecular structures, and proteomics data must be deposited in a publicly accessible database, and accession codes and associated hyperlinks must be provided in the “Data Availability” section.

Please include a “Data availability” subsection in the Online Methods. This section should inform readers about the availability of the data used to support the conclusions of your study, including accession codes to public repositories, references to source data that may be published alongside the paper, unique identifiers such as URLs to data repository entries, or data set DOIs, and any other statement about data availability. At a minimum, you should include the following statement: “The data that support the findings of this study are available from the corresponding author upon request”, describing which data is available upon request and mentioning any restrictions on availability. If DOIs are provided, please include these in the Reference list (authors, title, publisher (repository name), identifier, year). For more guidance on how to write this section please see: <http://www.nature.com/authors/policies/data/data-availability-statements-data-citations.pdf>

CODE AVAILABILITY

Please include a “Code Availability” subsection in the Online Methods which details how your custom code is made available. Only in rare cases (where code is not central to the main conclusions of the paper) is the statement “available upon request” allowed (and reasons should be specified).

For more information on our code sharing policy and requirements, please see: <https://www.nature.com/nature-research/editorial-policies/reporting-standards#availability-of-computer-code>

MATERIALS AVAILABILITY

As a condition of publication in Nature Methods, authors are required to make unique materials

promptly available to others without undue qualifications.

ORCID

Sincerely,
Rita

Rita Strack, Ph.D.
Senior Editor
Nature Methods

Reviewers' Comments:

Reviewer #1:

Remarks to the Author:

One critical issue in live-cell microscopy is the fast recording of large field-of-views with high optical sectioning capabilities. One remedy is light-sheet microscopy, where the thickness of the illuminating light-sheet determines the optical sectioning. In this work, Boden et al combine super-resolution microscopy based on the RESOLFT principle employing reversibly photoswitchable fluorescent proteins (rsFPs) with oblique plane microscopy to create effective fluorescence sheets of improved spatial thickness <250nm. They exemplify their approach on several cell-biological examples.

This is an impressive work with very nice results. In principle, it deserves publication in a highly visible journal like Nature Methods and I would be very supportive. However, it is a very special approach employing a complex custom-built instrument, which at the end might only find its way into very dedicated labs. Therefore, a more specialized journal (with respect to microscope instrumentation developments) might be more appropriate – happy to discuss this further.

Specific comments on the manuscript:

There are still some formatting errors (e.g. line 34) and the references are completely out of order (especially after reference 13).

I got confused which rsFPs are more appropriate for this approach, positive or negative switchers? Please elaborate a little bit more on this.

A little bit more details on how fast the images were/can be recorded should be given.

What is the dependency of the performance on experimental parameters like tilting angle, periodicity etc?

Line 81: Give more details on “if matched correctly”.

The lens-free scanning system is superb. Has this already been published (give reference)? Otherwise, stress this outstanding advancement more in the main text.

Line 117: How has the sectioning been measured here?

Line 147 and general: Be more precise with the spatial resolution in all the experiments. The number 250nm somehow comes out of the sky here.

Reviewer #2:

Remarks to the Author:

In this paper, the authors present multi-sheet reversible saturable optical fluorescence transitions (RESOLFT) microscopy. The idea of using reversibly switchable fluorescent proteins (RSFPs) to achieve super thin sections is new and has great potential for high-resolution light sheet microscopic imaging. Globally, the structure of the manuscript is clear, and the method is well demonstrated with both simulation and experiments. The proposed technique is interesting for wide audience from many different fields. In my opinion, this manuscript can be published in Nature Methods, given the following issues, as well as other reviewers’ concerns, are well addressed.

1. The use of reversibly switchable fluorescent proteins (RSFPs) is key to achieve super thin sections. The off-switching response of RSFPs to the 488-nm laser at different intensities, and the robustness of RSFPs versus the cycles of on-off switching, should be carefully explored and added in the supplementary file.
2. It is useful to characterize in detail the 3D structured patterns of the off-switching light and the 3D distribution of the read-out beam across the whole field of view (e.g., $100 \times 80 \times 15 \mu\text{m}^3$). Perhaps, also overlaying the two in different colors would be useful. I am wondering how well the read-out beam match with the off-switching patterns.
3. What are the factors that limit the performance of the proposed technique, for instance, imaging range ($100 \times 80 \times 15 \mu\text{m}^3$), speed (1-2Hz), spatial resolution (250 nm)?
4. I did not see a moving stage in the setup (Fig.1e), although I guess a moving stage might be necessary to image different sections with the camera.
5. Deskew and Richardson-Lucy deconvolution were used in the image analysis. These methods will show apparently higher spatial resolution than the value limited by the hardware. It is better to show the raw images without using these deconvolution approaches so that the readers can recognize the

pure contribution (the merit) of the proposed technique.

6. The details of the imaging procedure and figures 1 and 2 should be given, such as the numbers of axial sections in Fig. 2, the exposure time for each frame, etc.

Reviewer #3:

The authors of this manuscript present a super-sectioning microscopy technique for fast imaging the whole cells with a sectioning thickness of up to 10 folds better than that of traditional light sheet microscopy, in which the reversibly switchable fluorescent proteins (RSFPs) are used to generate multiple confined fluorescence-emitting sheets with the width of smaller than the width of the traditional sheet. The RSFPs are switched into on state using a widefield illumination of an on-switching light pulse, and then the interference fringes of off-switching light pulse switch the RSFPs from the on-state to the off-state, thus multiple confined fluorescence-emitting sheets are produced near the lowest intensity positions of the off-switching fringes, the width of which can be very small, exceeding the diffraction limit, by changing the time of intensity of the off-switching illumination. The effective fluorescence of the confined sheets is sequentially excited using a sheet of the read-out light pulse with a width of traditional light sheet. This work is a very important progress in the field of super-resolution microscopy and has significantly innovation. I think that this manuscript is suitable for publication in Nature Methods after answering the following questions. Please see below my concerns:

- 1) Switching the RSFPs among the different states is a key to reach super-sectioning imaging. Do these light pulses with the different functions require precise timing control? Please provide a detailed timing chart.
- 2) Please provide a detailed explanation of Fig. 1(c).
- 3) In Fig. 1(f), to show the off and read-out illumination patterns, 200nm fluorescent beads were used. Why do you choose 200nm fluorescent beads instead of fluorescent solutions?
- 4) In your imaging system, how is the effective pixel size defined? Will the size affect the axial resolution of your imaging system?
- 5) To better show sectioning ability of your system, a volume sample composed of dense fluorescent beads should be imaged. The axial resolution is determined by measuring the distance between two adjacent beads rather than measuring single small sample.

Author Rebuttal to Initial comments

NMETH-BC53143

We thank the reviewers for their feedback, which improved the clarity of the manuscript. Please find below a point by point answers to your questions.

Reviewer #1:

Remarks to the Author:

One critical issue in live-cell microscopy is the fast recording of large field-of-views with high optical sectioning capabilities. One remedy is light-sheet microscopy, where the thickness of the illuminating light-sheet determines the optical sectioning. In this work, Boden et al combine super-resolution microscopy based on the RESOLFT principle employing reversibly photoswitchable fluorescent proteins (rsFPs) with oblique plane microscopy to create effective fluorescence sheets of improved spatial thickness <250nm. They exemplify their approach on several cell-biological examples.

This is an impressive work with very nice results. In principle, it deserves publication in a highly visible journal like Nature Methods and I would be very supportive. However, it is a very special approach employing a complex custom-built instrument, which at the end might only find its way into very dedicated labs. Therefore, a more specialized journal (with respect to microscope instrumentation developments) might be more appropriate – happy to discuss this further.

We would first like to thank the reviewer for supporting our work and its relevance to the scientific community. We also sympathise with the reviewer's comment that the developed method constitutes a *special approach employing a complex custom-built instrument*. We would, however, like to emphasise that the concept of oblique plane light sheet microscopy using single primary objectives has quickly gained interest in the field with more than 20-30 research groups worldwide having adopted the idea (<https://amsikking.github.io/snoutclub/>) and a commercial version already launched by ASI/Leica (<https://www.asiimaging.com/products/light-sheet-microscopy/single-objective-light-sheet/>). From a hardware aspect, our system constitutes an add-on to these systems composed solely of standard optical components and a $0/\pi$ phase grating (the phase grating could also be substituted with any other optical assembly that creates two coherent beams of equal intensity) and standard diode lasers. As with all super-resolution endeavours, it is important to ensure that the end justifies the means for the specific application at hand. However, we believe that the Multi-sheet RESOLFT approach offers a significant and valuable improvement in image quality for a relatively small price in system complexity compared to diffraction-limited oblique plane microscopes. Additionally, the minimal requirement in on-off cycles on RSFPs compared to other RESOLFT implementations makes Multi-sheet RESOLFT compatible with a larger palette of photoswitchers, further decreasing the initial barriers in approaching the new method. To further facilitate the dissemination of this technology, all software used for the system is available and we have added a dedicated section in the supplementary information (Supplementary Note 4) to help potential adopters with optical design and alignment.

Below follows a point-by-point response to the specific issues raised by the reviewer.

Specific comments on the manuscript:

There are still some formatting errors (e.g. line 34) and the references are completely out of order (especially after reference 13).

We thank the reviewer for pointing this out and have addressed these issues by reformatting the citations and bibliography.

I got confused which rsFPs are more appropriate for this approach, positive or negative switchers? Please elaborate a little bit more on this.

NMETH-BC53143

We have revised these sections of the (lines 87-96) text and believe that it is now made clear how different types of RSFPs may be used in the system.

A little bit more details on how fast the images were/can be recorded should be given.

We have revised these sections of the text (lines 110-121) and believe that it is now made clear what imaging times were - and could hope to be - achieved.

What is the dependency of the performance on experimental parameters like tilting angle, periodicity etc?

For all the data acquired and presented, a tilting angle of 35 degrees is used. This choice has several reasons behind it. Firstly, it is important to note that the High-NA primary objective has a certain reachable 'angular range', where the maximum angle is defined by the NA as $\alpha_{max} = \arcsin(\frac{NA}{n})$ which for an NA of 1.35 gives ~73 degrees. This means that, in theory, by moving a focused point around on the back focal plane, a collimated beam can be generated out from the objective at any angle between 0 and 73 degrees with respect to the optical axis. Our performance tests, however, have shown that this range is highly optimistic if the output beam has a required diameter and that in practice, the output beam starts to degrade significantly after an angle of about 65 degrees. We have therefore limited our design choices to use only output beams with angles between 0 and 65 degrees for the generation of the read-out light sheet and off-switching interference pattern. For a given choice of the overall tilt α_{tilt} of the system, this leads to a resulting maximum angle around the tilt angle that can be used to create the converging beams for the interference pattern and the focusing of the light sheet, given by $\alpha_{residual} = \alpha_{max} - \alpha_{tilt}$. The minimum achievable periodicity of the interference pattern and the minimum thickness of the read-out light sheet is thus strictly dependent on the tilt angle chosen. Apart from affecting the periodicities and light sheet thicknesses that can be reached, the tilt angle also affects the quality of the detection. As the tilt angle increases, the optical quality of the detection starts to slowly deteriorate as not all the rays can be collected by the tertiary objective. In short, a larger tilt angle gives more flexibility in pattern generation, but a smaller tilt angle improves the quality of the optical detection. The choice of 35 degrees as the tilt angle allows a periodicity of 1.2 μm to be reached in the interference pattern which, with a matched light sheet thickness, gives a light sheet that will propagate with maintained peak intensity over at least 20-30 μm (FWHM along propagation direction of 28 μm), covering most cultured cells. If a larger tilt angle was used, an interference pattern with even smaller periodicity could be created. This would allow a sharper effective sheet confinement to be reached with lower illumination intensities (due to the steeper intensity gradient around the zero) and a potentially faster imaging thanks to a fewer number of cycles required, but would introduce several issues. Firstly, if matched in the same way with the read-out light sheet, it would require a thinner light sheet to be created, which would have a shorter propagation length and would result in a thinner sample section that could be imaged at high quality. Additionally, it would cause a deteriorated optical detection with a more asymmetrical optical PSF. As the reviewer clearly touches on, there are possibilities to tune these parameters and reach different trade-offs between speed, image quality and fatigue resistance. Using a higher-NA oil objective and utilizing the refraction induced at the coverslip interface could allow for improvements by increasing the angular range, but would come at the cost of system and alignment complexity and limited imaging possibilities deeper into the sample. Such a design would also likely include using a read-out light sheet that is thicker than the previously mentioned 'matching' would call for, which in turn would require a more advanced postprocessing algorithm to handle the crosstalk arising from exciting multiple confined sheets at the same time. A large and comprehensive exploration of all the possible configuration tweaks lies outside the scope of this proof-of-principle demonstration but may contribute to future optimizations of the technology and help in tailoring it to specific applications.

Line 81: Give more details on "if matched correctly".

NMETH-BC53143

We have rephrased that part of the text (lines 80-86) and hope that it is now clearer that “if matched correctly” refers to matching the periodicity of the off-switching with the width of the read-out sheet so that the confined sheets that are adjacent to the one being read out overlap with the zero-intensity regions of the airy shaped cross-section of the read-out light sheet.

The lens-free scanning system is superb. Has this already been published (give eference)?

Otherwise, stress this outstanding advancement more in the main text.

We sincerely appreciate the reviewer's enthusiasm about the lens-free scanning systems. The galvo-based system was shared publicly after its conception in a Twitter thread (tweetorial) and on Zenodo (DOI: 10.5281/zenodo.3653386). We rephrased the section (lines 122-127) in the main text to further stress the benefits of this assembly. Regarding the rotation stage assembly used for alignment (Supplementary Fig 2b), we are not aware of this alignment system being used previously but we hope that it can be of use to other builders in the future.

Line 117: How has the sectioning been measured here?

The sectioning capability of the standard diffraction-limited OPM comes solely from the thickness of the read-out light sheet. This thickness is determined by the focusing angle of the light sheet (effective NA) and is in our case estimated to be between $\text{FWHM} = 1-1.5 \mu\text{m}$ (we have changed in the text from 'around $1.5 \mu\text{m}$ ' to 'between $1-1.5 \mu\text{m}$ ', line 146, as it can be tuned by changing the size of the line on the back focal plane). The width of the light sheet in our system is estimated using simulations of interference models and experimental measurements using bead scans which both give a FWHM thickness of $\sim 1.1 \mu\text{m}$ (Supplementary Fig. 9c).

The sectioning ability when the Multi-sheet RESOLFT mode is used is addressed with both calculation and experimental validation. The estimations of achieved confinement are done with simulations/calculations based on the measured photophysical properties of the RSFPs. Fig 1b shows the expected confinement for different off-switching times with rsEGFP2 labelling. Corresponding curves can be calculated using other labelling proteins. The estimated confinement was then corroborated by experimentally measured line profiles in the images of the tubulin network and the Gag protein clusters shown in Fig 2a and Supplementary Fig. 10.

Line 147 and general: Be more precise with the spatial resolution in all the experiments.

The number 250nm somehow comes out of the sky here.

The reviewer raises a relevant point. We have revised the main text (lines 147-152) to address this and have taken care to elaborate on the resolution with additional quantification (Supplementary Fig. 10).

Reviewer #2:

Remarks to the Author:

In this paper, the authors present multi-sheet reversible saturable optical fluorescence transitions (RESOLFT) microscopy. The idea of using reversibly switchable fluorescent proteins (RSFPs) to achieve super thin sections is new and has great potential for high-resolution light sheet microscopic imaging. Globally, the structure of the manuscript is clear, and the method is well demonstrated with both simulation and experiments. The proposed technique is interesting for wide audience from many different fields. In my opinion, this manuscript can be published in Nature Methods, given the following issues, as well as other reviewers' concerns, are well addressed.

NMETH-BC53143

We appreciate the reviewers' endorsement of the proposed method and the conditioned support for the publication of the manuscript in Nature Methods. We have carefully reviewed the reviewer's issues and addressed them in a point-by-point manner below.

1. The use of reversibly switchable fluorescent proteins (RSFPs) is key to achieve super thin sections. The off-switching response of RSFPs to the 488-nm laser at different intensities, and the robustness of RSFPs versus the cycles of on-off switching, should be carefully explored and added in the supplementary file.

We sympathize with the reviewer's comment on the importance of carefully characterizing the photophysical response of the RSFPs to different illumination intensities and their limitations with respect to repetitive cycling. As suggested, we have added a dedicated section in the supplementary (Supplementary Note 1 and Supplementary Figure 1) to address these issues in the context of the proposed method.

2. It is useful to characterize in detail the 3D structured patterns of the off-switching light and the 3D distribution of the read-out beam across the whole field of view (e.g., 1008015 m3). Perhaps, also overlaying the two in different colors would be useful. I am wondering how well the read-out beam match with the off-switching patterns.

As pointed out by the reviewer, accurate alignment between the different illuminations in each step of the imaging sequence is essential for the performance of the system. Achieving accurate alignment relies on three main parameters: (i) The static intensity patterns of the different illuminations (the periodicity of the off-pattern, the tilt angle of both off-pattern and light-sheet and lateral co-alignment of light-sheet with zero of off-pattern), (ii) the rotational co-alignment between the off-pattern and the read-out sheet, and (iii) the exact matching of the scanning sequence (galvo motion) with the said pattern periodicity. During alignment, the three properties are calibrated in different steps. To clarify this procedure and demonstrate the accuracy/limitations we have added a section in the supplementary notes and an additional supplementary figure elaborating on these aspects (Supplementary Note 4 and Supplementary Fig. 8).

3. What are the factors that limit the performance of the proposed technique, for instance, imaging range (1008015 m3), speed (1-2Hz), spatial resolution (250 nm)?

Field of view: The limitations on the size of the field of view (FOV) are set by different factors for different dimensions (x,y,z).

For the first lateral dimension (x) which is not the scanning dimension, the limit is set by the smallest of the (i) optical field of view of the detection (usually determined by the field of view of the primary objective), (ii) the size (in the x-direction) of the area in which the illumination patterns exhibit sufficiently homogenous intensity, or (iii) in rare cases the size of the camera sensor used. In our case, the width of the homogeneous part of the off-pattern corresponds roughly to the size of the diffraction-limited field of view of the primary objective at about 100 μm . This sets the limit for the final FOV in the x-direction.

For the second lateral dimension (y), corresponding to the scan direction, the limit is set by the smallest of (i) the optical field of view of the detection and (ii) the size (in the y-direction) of the area in which the off-pattern exhibits sufficiently homogenous intensity and (iii) the length of scanning that can be accommodated by the system. In our setup, the lens-free scanning module can only accommodate a scan length of around $\pm 30 \mu\text{m}$ from the centre position which sets the FOV limit in this dimension.

For the axial dimension (z), the achievable depth depends on how long sheets can be created and efficiently read out along the direction of propagation of the read-out light sheet. This in turn depends on how long the read-out sheet maintains sufficiently strong intensity to excite enough fluorescence from the central confined sheet and also keeps a sufficiently thin profile so as to not

NMETH-BC53143

excite too much fluorescence from adjacent confined sheets and also have. Simulations give a good estimation of the light sheet profile along the propagation direction, and measurements set the FWHM of the intensity in the central sheet lobe along the propagation direction at between 25-30 μm . Scaling this length with a factor of $\sin(\alpha)$ gives the corresponding depth along the z-direction due to the tilt angle α of the sheet and equals between 14-17 μm . Simply choosing the FWHM of the central intensity as a measure for the length of the effective sheet clearly does not capture all the nuances that the discussion on this matter could encompass. The limit on imaging depth is not strict but rather the quality will gradually degrade as imaging depth increases. We however believe, based on the above reasoning and experimental results, that an imaging depth of 15 μm along the z-direction is achievable without significant loss in image quality. We have added a supplementary figure (Supplementary Fig. 9) to illustrate the relationship between light sheet propagation and imaging depth and to quantify the light sheet thickness.

Speed

Several different factors influence the imaging speed. Between each read-out sweep, the spatial on-state pattern needs to be imprinted to the sample volume. This is done using an on-switching pulse and consecutive off-switching pulse. This switching cycle needs to be performed 10-20 times per volumetric recording and each switching cycle takes between 35-50 ms for rsEGFP2 and 100-220 ms for rsEGFP(N205S). This time is currently limited by the achievable power density of the switching illuminations and could potentially be drastically reduced. Other RESOLFT implementations have demonstrated that the switching can be achieved on sub-millisecond time scales if sufficient illumination power is available. The other large component of the total imaging time is the read-out sweep. For the herein-demonstrated volumes, between 20 and 40 planes are read out each sweep, and each plane usually takes 1-2 ms to excite and read out the camera frame. This time is currently mainly limited by the speed of the camera (minimum exposure time and read-out time), but could potentially be reduced given as the speed of state-of-the-art cameras continues to improve. In a potential future scenario where switching time on the order of 1 ms can be achieved and the camera speeds increase. It is not unfeasible that volumetric imaging speeds of >5 Hz could be reached for whole-cell imaging.

Spatial resolution

The spatial resolution of the Multi-sheet RESOLFT system is dependent on two distinctly separate parameters. Firstly, the super-sectioning property of the MS-RESOLFT systems improves the resolution along the direction orthogonal to the tilted detection plane. In this direction, there is no conceptual limit to what effective confinement can be reached, as shown in Fig. 1b. Practically however, we find that it is hard to push the confinement significantly beyond ~ 150 nm. The exact interplay of parameters that give rise to this limit is still elusive, but we believe it is a combination of several factors. (i) With the current pattern periodicity and illumination power, reaching a significantly thinner effective sheet would require very long off-switching times (hundreds of ms) which could mean that the confinement of the sheets is limited by the motion of the RSFPs in the sample (moving RSFPs would all be switched off as they move through the off-switching pattern). (ii) Minor imperfections/aberration in the optical path (possibly induced by the sample) may cause the zero-intensity regions of the interference pattern to not be completely zero, which would mean that for long off-switching times, even RSFPs residing in the intensity-minima would start switching off, killing the overall image signal. (iii) Even with perfect intensity zeros, increasingly sharp confinement will inevitably decrease the SNR of the final data, unless the sampling distance is decreased correspondingly. This itself poses a practical limit on sheet confinement as too dense sampling is detrimental for imaging time and the possibility for longer time-lapse recordings.

Concerning the ‘in-plane’ resolution of the tilted acquisition, i.e. the resolution within the plane that is parallel to the tilted detection plane, is still diffraction limited, since the RESOLFT improvement is purely achieved orthogonally to this plane. Thus, improving this resolution would require a conceptual addition on top of the MS-RESOLFT, either by the addition of switching illumination patterns with high frequencies in other dimensions, or by coupling the MS-RESOLFT system with other types of super-resolution approaches.

NMETH-BC53143

4. I did not see a moving stage in the setup (Fig.1e), although I guess a moving stage might be necessary to image different sections with the camera.

We thank the reviewer for kindly pointing this out. Indeed, the sample is placed on a 3-Axis NanoMax Piezo Stage with Stepper Motor Actuators for positioning the sample. However, neither the piezo nor the motorized stage has any role in the actual acquisition sequence because as we scan both the light sheet (across the FOV, hopping each 1,2 μm) and the interference pattern (across the periodicity of 1,2 μm with steps of 105-210 nm) using the galvo mirror. We have added a mention of the motorized stage in the description of the setup (lines 232-233).

5. Deskew and Richardson-Lucy deconvolution were used in the image analysis. These methods will show apparently higher spatial resolution than the value limited by the hardware. It is better to show the raw images without using these deconvolution approaches so that the readers can recognize the pure contribution (the merit) of the proposed technique.

We agree with the reviewer that deconvolution approaches may yield images that appear to have higher resolution (contain higher frequencies) than the imaging system used to collect the raw data actually contains. For this reason, we have chosen to perform all quantitative measurements that support our claims of sheet confinement of the non-deconvolved data. To further increase transparency, we have added the non-deconvolved versions of all the presented data in Supplementary Figures 4 and 5.

6. The details of the imaging procedure and figures 1 and 2 should be given, such as the numbers of axial sections in Fig.2, the exposure time for each frame, etc.

We have revised the manuscript so that this point is further clarified by moving some information from the supplementary material to the main text (lines 142, 157, 174) and pointing the reader to the supplementary information where needed (lines 118-119 and 179-180).

Reviewer #3

The authors of this manuscript present a super-sectioning microscopy technique for fast imaging the whole cells with a sectioning thickness of up to 10 folds better than that of traditional light sheet microscopy, in which the reversibly switchable fluorescent proteins (RSFPs) are used to generate multiple confined fluorescence-emitting sheets with the width of smaller than the width of the traditional sheet. The RSFPs are switched into on state using a widefield illumination of an on-switching light pulse, and then the interference fringes of off-switching light pulse switch the RSFPs from the on-state to the off-state, thus multiple confined fluorescence-emitting sheets are produced near the lowest intensity positions of the off-switching fringes, the width of which can be very small, exceeding the diffraction limit, by changing the time of intensity of the off-switching illumination. The effective fluorescence of the confined sheets is sequentially excited using a sheet of the read-out light pulse with a width of traditional light sheet. This work is a very important progress in the field of super-resolution microscopy and has significantly innovation. I think that this manuscript is suitable for publication in Nature Methods after answering the following questions. Please see below my concerns:

We thank the reviewer for the effort put into refereeing our manuscript and for the support of publication in Nature Methods.

NMETH-BC53143

1) Switching the RSFPs among the different states is a key to reach super-sectioning imaging. Do these light pulses with the different functions require precise timing control? Please provide a detailed timing chart.

As the reviewer accurately points out, the timing of pulses is of great importance to achieve super-sectioning performance. On top of requiring precise control of the sequence of pulses, the length of each pulse is essential to achieve just the right amount of switching and/or excitation to optimize performance. The figure below illustrates the timing chart for an example acquisition using rsEGFP2 labelling. The full sequence consists of 10 cycles, each cycle consisting of an on-switching pulse (magenta), an off-switching pulse (darker blue), and a read-out sweep (lighter blue at the top or pulsed blue at the bottom).

2) Please provide a detailed explanation of Fig. 1(c).

Fig 1(c) illustrates the imaging scheme by showing three time points within the full imaging sequence. The green tilted planes illustrate which parts of the volumetric sample have been imaged at the three different time points. The leftmost image illustrates the planes that have been acquired after the first switching and read-out cycle (C_1). Here, only a single 'set' of planes has been confined and read out, distances $1.2 \mu\text{m}$ apart as determined by the periodicity of the off-switching interference pattern. The centre image illustrates the situation after a few (n) on-off and read-out cycles (C_n). The repeated read-out cycles are performed identically to each other, with the only difference being that the relative position of all the illuminations is shifted slightly (usually 105 or 210 nm) with respect to the sample. In this way, each on-off and read-out cycle probes a slightly different set of confined planes in the sample, gradually filling up the non-probed space between the initially acquired planes. In the center image, roughly half the total number of planes have been acquired. The rightmost image shows the situation after the final cycle (C_N) has been acquired. Here, the illustration shows that a homogeneous sampling of the whole volume is achieved at the end.

3) In Fig.1(f), to show the off and read-out illumination patterns, 200nm fluorescent beads were used. Why do you choose 200nm fluorescent beads instead of fluorescent solutions?

Fig. 1(f) is acquired using a coverslip onto which a very small number of bright 200 nm fluorescent beads have been added and adhered to the coverslip surface. The beads are so sparse that a small central FOV on the camera can readily be cropped containing only a single bead. We then illuminate with the different illumination patterns while scanning the single fluorescent bead in a raster scanning manner through the illumination pattern and acquiring a camera frame at every position. By measuring

NMETH-BC53143

the total fluorescence emitted from the bead and captured by the camera at every position we can then plot an intensity map of the different illumination patterns and inspect their alignment with respect to each other. We have developed this protocol as an accurate and robust way of probing the three-dimensional light patterns and do not believe we could achieve a similar measurement using a fluorescent solution.

4) In your imaging system, how is the effective pixel size defined? Will the size affect the axial resolution of your imaging system?

Due to the tilted geometry of the detection, discussions about the sampling become slightly more complex, since the acquired data points do not form a regularly spaced cartesian grid in 3D. The distances between the sample points in 3D are defined by the effective pixel size of the camera together with the tilt of the detection system and the spacing between adjacent acquired planes.

The effective pixel size of the camera capturing the raw data is currently 116 nm, this pixel size satisfies the Shannon-Nyquist criteria for the lateral optical resolution of tilted detection, which is estimated to be ~250 nm, while minimizing the noise and read-out time of the camera. The distance between adjacent acquired planes depends on the tilt of the system (α_{tilt} is always 35 degrees in our setup) and the lateral scan step used during acquisition (d_{scan} which is 105 or 210 nm in the presented data). The shortest distance between two adjacent planes is then calculated as $d_{scan} * \sin(\alpha_{tilt})$ which results in either 60 or 120 nm. When a very thin sheet confinement is created resulting in sheets <200 nm thick, the smaller spacing is needed to fully satisfy the Shannon-Nyquist sampling criterion, while with more relaxed off-switching, the larger distance would suffice, and would also give a faster imaging. The size of the scanning step should thus be chosen to match the axial resolution expected of the system. Additionally, a smaller scanning step, and thus a smaller interplanar distance, will also increase the signal-to-noise ratio of the final image and can in that sense contribute to boosting the signal of higher frequencies and giving an apparent better image quality/resolution.

The pixel size in the final reconstructed volumetric images can be freely chosen in the reconstruction algorithm. Unless certain very harsh geometrical constraints are adhered to in the acquisition, the geometrical transformation required to reconstruct onto a traditional cartesian voxel volume inevitably requires resampling the data. This resampling can be done onto a voxelized volume with arbitrary voxel size. In practice, we choose to often reconstruct the data onto a voxelized grid of 75 or 100 nm voxels, which satisfies the Shannon-Nyquist sampling criterion for a resolution of 200-250 nm while resulting in manageable data sizes and computational times.

5) To better show sectioning ability of your system, a volume sample composed of dense fluorescent beads should be imaged. The axial resolution is determined by measuring the distance between two adjacent beads rather than measuring a single small sample.

The reviewer raises a relevant point regarding the quantification of the sectioning capability of our Multi-sheet RESOLFT system. As the reviewer also states, a rationally sound way to investigate this would be by imaging a sample composed only of small point-like emitters distributed over a three-dimensional volume. Unfortunately, a sample of purely point-like structures labelled with reversibly switchable fluorescent proteins is not trivially created. To provide a more in-depth characterization, we have instead more thoroughly dissected a dataset of Gag-IM-rsEGFP2 transfected HeLa cells. The Gag proteins labelled in the Gag-IM-rsEGFP2 transfection show a clear clustering tendency and can thus at some locations be used as nearly point-like emitters in three dimensions. In Supplementary Fig. 10 we have taken several different approaches to characterize the three-dimensional effective PSF and the axial sectioning capability of the Multi-sheet RESOLFT system. To more accurately quantify the effective PSF size and decrease the influence of spurious noise, the measurements are performed on data created by averaging five raw data acquisitions. The data has been reconstructed using the simple deskewing algorithm i.e. without any deconvolution.

We have identified Gag protein clusters in the Gag-IM-rsEGFP2 transfected HeLa that we consider to be nearly point-like emitters in 3D. Measurements were made on these structures in the

NMETH-BC53143

image along both the x, y, and z directions. The numbers are again within the expected range, showing sizes in the x-y plane corresponding to the resolution of the optical detection (~250 nm) and a slightly smaller size along the z-direction due to the Multi-sheet RESOLFT confinement.

As the reviewer also points out, another established way of measuring effective resolution is to extract the shortest distance between two adjacent emitters that can be distinguished in the image. For perfectly noiseless data, two emitters at distance D should be just distinguishable if the FWHM of the effective PSF is smaller than or equal to D (the Rayleigh criterion gives a ~20% intensity dip). It should however be stressed that the ability to reliably distinguish two emitters is also dependent on the noise of the image. For high-resolution microscope images, which are never noise-free, a distance slightly larger than the FWHM of the PSF is generally needed to reliably distinguish two emitters. In Supplementary Fig. 10c, we have identified two instances where closely located structures are just distinguishable. Our measurements suggest again that the Multi-sheet RESOLFT system can reach axial resolutions of down to around 200 nm, as structures distanced just above this can be distinguished in the experimental data.

Decision Letter, first revision:

Dear Ilaria,

Thank you for submitting your revised manuscript "Super-sectioning with Multi-sheet REversible Saturable Optical Fluorescence Transitions (RESOLFT) microscopy" (NMETH-A53143A). It has now been seen by the original referees and their comments are below. The reviewers find that the paper has improved in revision, and therefore we'll be happy in principle to publish it in Nature Methods, pending minor revisions to comply with our editorial and formatting guidelines.

TRANSPARENT PEER REVIEW

ORCID

Sincerely,
Rita

Rita Strack, Ph.D.
Senior Editor

Nature Methods

Reviewer #1 (Remarks to the Author):

The authors have well responded to all of my previous concerns and revised the manuscript well accordingly. I suggest publication in Nature Methods.

One very minor thing. The authors have well answered to my previous question on the dependency of the performance on experimental parameters like tilting angle, periodicity etc, which they for good reasons have not included in the current manuscript. Maybe they could mention that this will be a topic of future work.

Reviewer #2 (Remarks to the Author):

The authors have clarified all my major concerns during the revision process, and therefore I agree the publication of this manuscript in nature methods.

Reviewer #3 (Remarks to the Author):

The authors responded well to all my concerns.

Author Rebuttal, first revision:

NMETH-A53143A

Reviewer #1:

Remarks to the Author:

The authors have well responded to all of my previous concerns and revised the manuscript well accordingly. I suggest publication in Nature Methods.

One very minor thing. The authors have well answered to my previous question on the dependency of the performance on experimental parameters like tilting angle, periodicity etc, which they for good reasons have not included in the current manuscript. Maybe they could mention that this will be a topic of future work.

We thank the reviewer for the support for publication. We have added a sentence stating that exploration of all parameters is a topic of future work.

Reviewer #2:

Remarks to the Author:

The authors have clarified all my major concerns during the revision process, and therefore I agree the publication of this manuscript in nature methods.

We thank the reviewer for the support for publication.

Reviewer #3:

Remarks to the Author:

The authors responded well to all my concerns.

We thank the reviewer for the support for publication.

Final Decision Letter:

Dear Ilaria,

I am pleased to inform you that your Article, "Super-sectioning with Multi-sheet REversible Saturable Optical Fluorescence Transitions (RESOLFT) microscopy", has now been accepted for publication in Nature Methods. The received and accepted dates will be July 11, 2023 and Jan, 24, 2024. This note is intended to let you know what to expect from us over the next month or so, and to let you know where to address any further questions.

Over the next few weeks, your paper will be copyedited to ensure that it conforms to Nature Methods style. Once your paper is typeset, you will receive an email with a link to choose the appropriate publishing options for your paper and our Author Services team will be in touch regarding any additional information that may be required. It is extremely important that you let us know now whether you will be difficult to contact over the next month. If this is the case, we ask that you send us the contact information (email, phone and fax) of someone who will be able to check the proofs and deal with any last-minute problems.

Please note that *Nature Methods* is a Transformative Journal (TJ). Authors may publish their research with us through the traditional subscription access route or make their paper immediately open access through payment of an article-processing charge (APC). Authors will not be required to make a final decision about access to their article until it has been accepted. [Find out more about Transformative Journals](https://www.springernature.com/gp/open-research/transformative-journals)

Authors may need to take specific actions to achieve [compliance with funder and institutional open access mandates](https://www.springernature.com/gp/open-research/funding/policy-compliance-faqs). If your research is supported by a funder that requires immediate open access (e.g. according to [Plan S principles](https://www.springernature.com/gp/open-research/plan-s-compliance)) then you should select the gold OA route, and we will direct you to the compliant route where possible. For authors selecting the subscription publication route, the journal's standard licensing terms will need to be accepted, including [self-archiving policies](https://www.springernature.com/gp/open-research/policies/journal-policies). Those licensing terms will supersede any other terms that the author or any third party may assert apply to any version of the manuscript.

If you are active on Twitter/X, please e-mail me your and your coauthors' handles so that we may tag you when the paper is published.

Best regards,
Rita

Rita Strack, Ph.D.
Senior Editor
Nature Methods